# Distinguishing artificial spin ice states using magnetoresistance effect for neuromorphic computing

Wenjie Hu[1,2], Zefeng Zhang[3,4], Yanghui Liao[1,2], Qiang Li [1,2], Yang Shi[1,2], Huanyu Zhang[1,2], Xumeng Zhang [3], Chang Niu[1,2], Yu Wu[1,2], Weichao Yu[1,5], Xiaodong Zhou [1,2,5], Hangwen Guo[1,2,5], Wenbin Wang[1,2,5], Jiang Xiao [1,2,5,6,7], Lifeng Yin [1,2,5,6,7,8] ✉, Qi Liu[3,8] ✉ & Jian Shen [1,2,5,6,7,9] ✉

Artificial spin ice (ASI) consisting patterned array of nano-magnets with frustrated dipolar interactions offers an excellent platform to study frustrated physics using direct imaging methods. Moreover, ASI often hosts a large number of nearly degenerated and non-volatile spin states that can be used for multi-bit data storage and neuromorphic computing. The realization of the device potential of ASI, however, critically relies on the capability of transport characterization of ASI, which has not been demonstrated so far. Using a tri-axial ASI system as the model system, we demonstrate that transport measurements can be used to distinguish the different spin states of the ASI system. Specifically, by fabricating a tri-layer structure consisting a permalloy base layer, a Cu spacer layer and the tri-axial ASI layer, we clearly resolve different spin states in the tri-axial ASI system using lateral transport measurements. We have further demonstrated that the tri-axial ASI system has all necessary required properties for reservoir computing, including rich spin configurations to store input signals, nonlinear response to input signals, and fading memory effect. The successful transport characterization of ASI opens up the prospect for novel device applications of ASI in multi-bit data storage and neuromorphic computing.

Artificial spin ice (ASI) is formed by patterned array of nano-magnets with strong and geometrically frustrated dipolar interaction[1,2]. Each nano-magnet behaves as a microscopic Ising spin whose orientation can be conveniently seen by direct magnetic imaging methods. This allows one to study frustration physics using microscopy tools. Interesting physical phenomena such as emergent magnetic monopoles[3–5], vertex-based frustration[6–8], phase transition[9,10], and chiral dynamics[11] have been observed and studied extensively in ASI networks. While the ground state of ASI follows Pauling's ice rule, the frustrated interactions often lead to numerous nearly degenerated states in ASI[5,12]. These states in ASI are nonvolatile and thus have been proposed for multi-bit data storage and neuromorphic computing applications[13–17].

[1]State Key Laboratory of Surface Physics, Institute for Nanoelectronic Devices and Quantum Computing, and Department of Physics, Fudan University, Shanghai, China. [2]Shanghai Qi Zhi Institute, Shanghai, China. [3]Frontier Institute of Chip and System, Fudan University, Shanghai, China. [4]Research Institute of Intelligent Complex Systems and ISTBI, Fudan University, Shanghai, China. [5]Zhangjiang Fudan International Innovation Center, Fudan University, Shanghai, China. [6]Shanghai Research Center for Quantum Sciences, Shanghai, China. [7]Collaborative Innovation Center of Advanced Microstructures, Nanjing, China. [8]State Key Laboratory of Integrated Chips and Systems, Fudan University, Shanghai, China. [9]Shanghai Branch, CAS Center for Excellence and Synergetic Innovation Center in Quantum Information and Quantum Physics, Shanghai, China. ✉e-mail: lifengyin@fudan.edu.cn; qi_liu@fudan.edu.cn; shenj5494@fudan.edu.cn

Besides magnetic imaging, the common characterization tools used for ASI include ferromagnetic resonance[18,19], resonant soft X-ray scattering[9,20], and X-ray photon correlation spectroscopy[21,22]. While these methods are powerful for distinguishing the nearly degenerated states and understanding the physics of ASI, they are not practical for the device applications of ASI. In order to fully realize the device potential of ASI, transport characterization is highly desired. It has been reported that magnetoresistance measurement of an ASI system with connected nanowires had unique signature of the ground state[23]. However, it is challenging to use transport measurements to distinguish the large number of nearly degenerated states with the subtle difference between their spin configurations[24,25]. So far, no transport experiment has been successfully performed to demonstrate the capability of distinguishing the nearly degenerated states in ASI.

In this paper, we use combined giant magnetoresistance[26] (GMR) and anisotropic magnetoresistance[27] (AMR) effects to distinguish the nearly degenerated states in a tri-axial ASI structure. The tri-axial ASI structure consists of three sublattices of nano-magnets oriented along horizontal, vertical, and diagonal directions in a square lattice, respectively. It has been demonstrated that eight different states with long-range ordered spin configurations can be achieved by applying an in-plane magnetic field with different angles[28,29]. In our device set-up, a tri-layer structure consisting of a permalloy (Py) base layer, a Cu spacer layer and the tri-axial ASI top layer is fabricated for lateral transport measurements. The eight spin states of the ASI structure exhibit eight distinct levels of resistance, indicating that different ASI spin states can clearly be distinguished using transport measurements based on our device structure. After including the GMR and AMR effects, our model calculation based on the resistance network show good consistency with experimental measured data. We have further demonstrated that the tri-axial ASI system has full capabilities for reservoir computing.

## Results and discussion

Figure 1a shows the schematic of the tri-layer structure grown on a Si/SiO$_2$ substrate for transport measurements. The Py base layer and the Cu spacer layer are 6 nm and 4 nm thick, respectively. The top tri-axial ASI layer is formed by 340,000 nano-magnets fabricated from a 16 nm thick Py layer by ion beam etching (see method section). Each nano-magnet has an elongated shape with 470 nm in length and 170 nm in width. Figure 1b shows the scanning electron microscopy (SEM) image of the tri-axial ASI pattern, in which the base unit is marked by the white box. Due to the interplay between the shape anisotropy of each

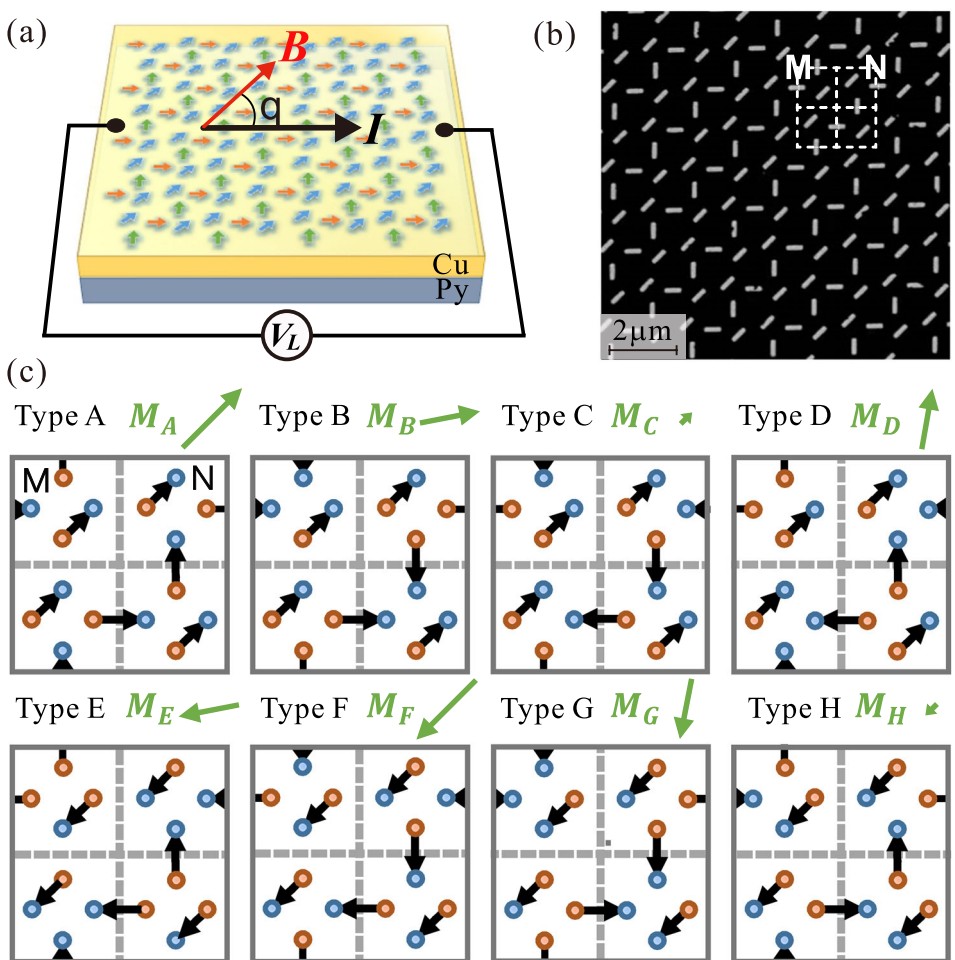

**Fig. 1 | Structure of tri-layer ASI device. a** Schematic diagram of the tri-layer structure for transport measurement. The top tri-axial ASI layer is fabricated on the Py base layer and the Cu spacer layer. The arrows in three colors represent the three axial sublattices. The black and red arrows indicate the directions of the applied current and the in-plane magnetic field. **b** Scanning electron microscopy image of tri-axial ASI layer. The size of each nano-magnet is 470 nm in length and 170 nm in width. The inset white dotted square is a base unit. And two type of inversed plaquettes are marked with M and N. **c** The eight possible spin states are named from Type A to Type H. The arrows are the orientation of the nano-magnets. And the colored circles are the magnetic charges at the ends of the nano-magnets (red for negative, blue for positive). The green arrows are the net magnetization for the corresponding spin states. The calculated total net magnetization for spin states type A to type H are $2\sqrt{2}+4M_s$ at 45°, $2\sqrt{6}M_s$ at 10°, $4-2\sqrt{2}M_s$ at 45°, $2\sqrt{6}M_s$ at 80°, $2\sqrt{6}M_s$ at 190°, $2\sqrt{2}+4M_s$ at 225°, $2\sqrt{6}M_s$ at 260°, $4-2\sqrt{2}M_s$ at 225°, respectively ($M_s$ is the magnetization of a single nano-magnet).

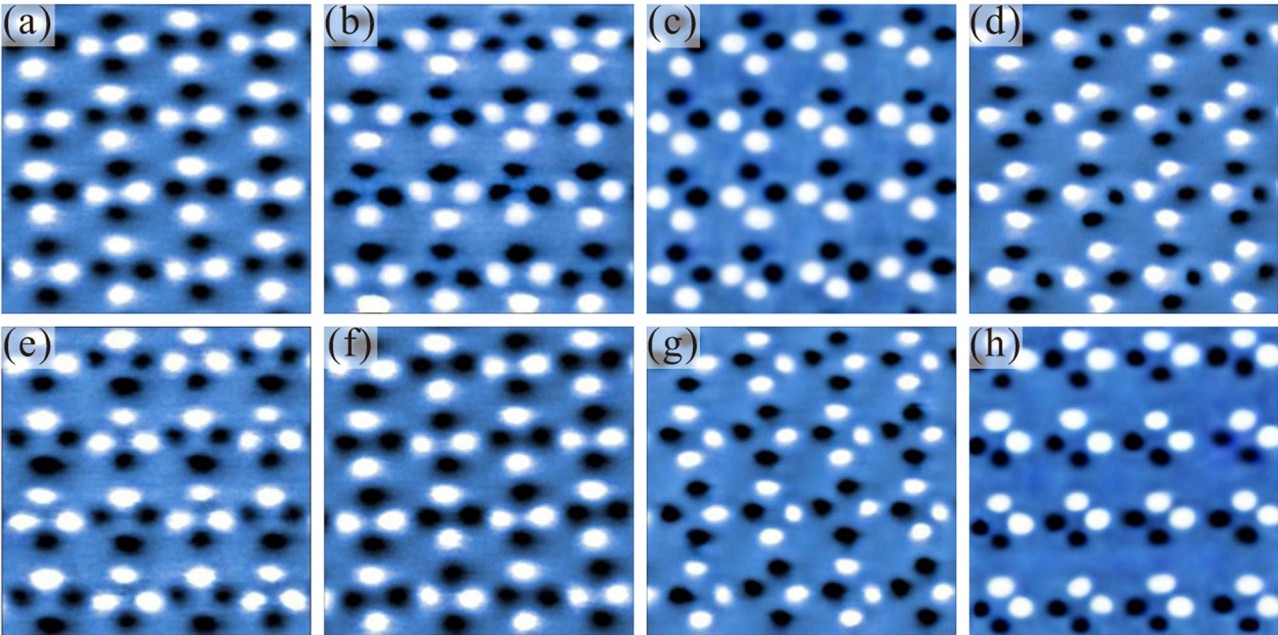

**Fig. 2 | Magnetic force microscopy images of the eight long-range ordered spin configurations. a–h** is corresponding to the Type A to Type H spin states in Fig. 1c. The eight states are created by the sequence of the applied magnetic field from $\theta = 45°$, 292°, 158°, 112°, 202°, 248°, 338° to 112°.

nano-magnet and the frustrated dipolar interaction between the nano-magnets, there are eight nearly degenerated spin states with long-ranged order in the tri-axial ASI[28], which can be prepared by external in-plane magnetic field applied in designed angle and sequence. Figure 1c shows the schematic of the eight spin states for a base unit denoted as Type A to Type H. Since the magnetic flux at the two ends of each dipole has opposite directions, we can treat them as "in" and "out" ends. The "in" and "out" ends form a square, and can be distinguished as three types: 1) "2 in-2 out-D" with two degenerated states (A and F), where the two "in" ends are positioned in diagonal axis; 2) "2 in-2 out-S" with degenerated states (C and H), where one "in" and one "out" ends are positioned in diagonal axis; 3) "1 in-3 out" (D and E) and "3 in-1 out" (B and G), where the four states are degenerated.

The eight long-range ordered spin states have been confirmed by magnetic force microscopy (MFM) images as shown in Fig. 2. Images **a–h** show expected spin configurations corresponding to the Type A to Type H spin states in Fig. 1c. From a demagnetized state (see supplementary Fig. S1), the eight states are created by subsequent application of field (-80 Oe) along $\theta = 45°$, 292°, 158°, 112°, 202°, 248°, 338°, and 112°, where $\theta$ represents the angle between the field direction and the horizontal direction as indicated in Fig. 1a. Once formed, the eight states are stable after removal of the magnetic field. We note that spin state Type A can be reobtained from spin state Type H by applying the magnetic field along $\theta = 45°$.

The formation of the eight spin states can be simulated by micromagnetic modeling which yields more detailed spin structures of the tri-layer system. In our simulation, the thickness of the Py base layer and the Cu spacer layer are the same as those of the experiments, while the ASI top layer is simplified to contain one repeatable base unit formed by 8 nano-magnets with the same experimental dimension. Similar with the experimental measurements, in the simulation a 120 Oe in-plane magnetic field is applied subsequently at an angle $\theta = 45°$, 292°, 158°, 112°, 202°, 248°, 338°, and 112°. After each ASI spin state is formed, a 20 Oe in-plane field is applied along $\theta = 0°$ to ensure the magnetization of the Py base layer is along $\theta = 0°$ direction without affecting the formed ASI spin state. By using the COMSOL multi-physics finite element code[30], we obtain eight different spin configurations of the nano-magnets in the ASI top layer and Py base layer (see

supplementary Fig. S2). Figure 3 shows simulated spin structures of the top ASI base unit and the corresponding regions of the Py layer for two of the eight spin states, i.e., Type A and Type F. Note that the spin direction of the two ends of each nano-magnet curls away from the long axis due to both dipolar interaction between the nano-magnets and the minimization of the demagnetizing energy of each nano-magnet[31,32]. The dipolar field generated by the non-colinear spin structure of each nano-magnet will affect the local spin alignments in the Py base layer, especially in regions near the ends of the nano-magnets. The non-colinear spin structures of the ASI nano-magnets and the Py base layer break the degeneracy of spin configurations in space distribution and allows us to distinguish different spin states from transport measurement.

Although the eight spin states can be distinguished by MFM, it is not practical for device applications. The calculated total net magnetization for spin states type A to type H are $2\sqrt{2} + 4M_s$ at 45°, $2\sqrt{6}M_s$ at 10°, $4 - 2\sqrt{2}M_s$ at 45°, $2\sqrt{6}M_s$ at 80°, $2\sqrt{6}M_s$ at 190°, $2\sqrt{2} + 4M_s$ at 225°, $2\sqrt{6}M_s$ at 260°, $4 - 2\sqrt{2}M_s$ at 225°, respectively ($M_s$ is the magnetization of a single nano-magnet), which are too small to be characterized by any magnetometry measurements. We thus attempt to distinguish them using transport measurements[33,34], which is highly desirable for device applications. Figure 4 shows the room temperature measured resistance of the eight spin states in the ASI tri-layer structure shown in Fig. 1a. Prior to and upon the transport measurements of a particular spin state, a small in-plane field (-10 Oe) was applied to the tri-layer structure along horizontal direction ($\theta = 0°$), which is big enough to ensure the alignment of the magnetization of the Py base layer along the current direction but is small enough not to affect the ASI spin state. The fixed magnetization direction of the Py base upon each transport measurement eliminates the influence of the AMR effect from the Py base layer caused by the switching field. As shown in Fig. 4a, the resistance of the eight ASI spin states are clearly distinguishable, which is better demonstrated by the separated eight peaks of the histogram of the acquired resistance data (Fig. 4b). We note here that the fluctuation of the resistance during the measurements is smaller than 0.01395 Ω, which is well below the resistance difference between any of the two spin states. The stability of these spin states have also been tested

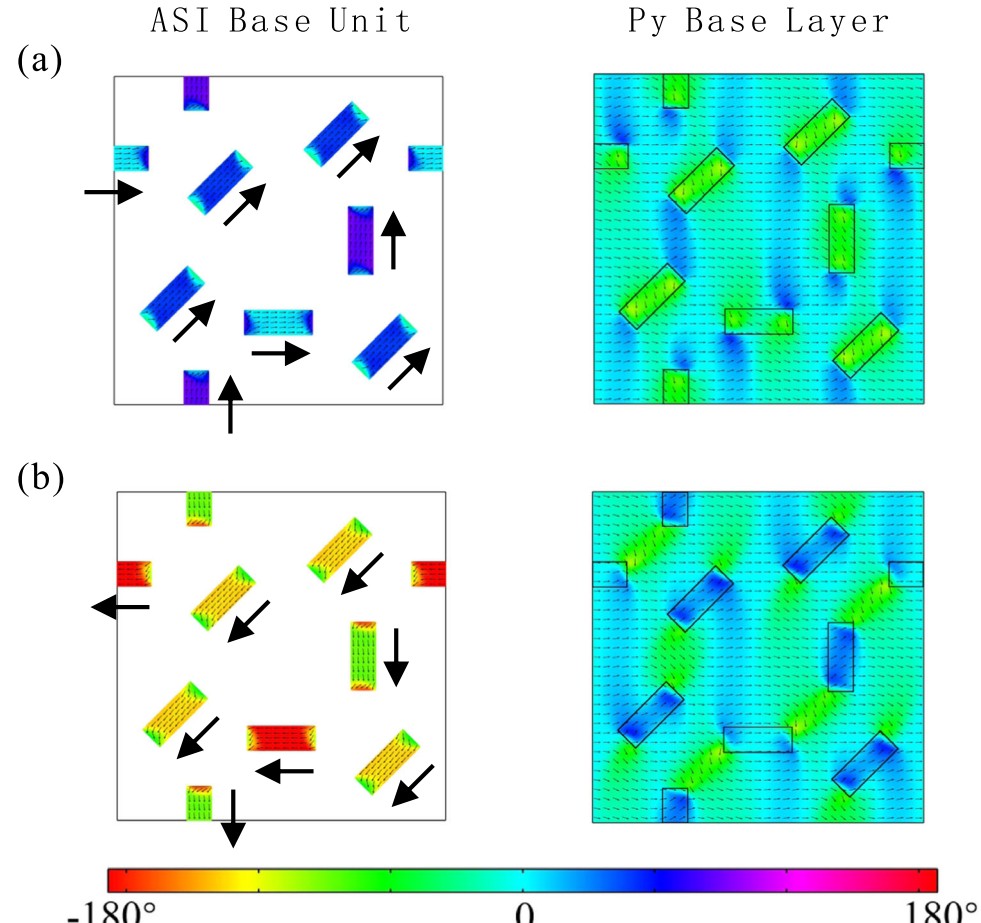

**Fig. 3 | Detailed spin structure of the tri-layer ASI device simulated by micromagnetic modeling. a, b** are the Type A and Type F spin states in Fig. 1c respectively. The left and right images represent the simulated spin structures of the ASI base unit and the base Py layer, respectively. The color code represents the direction of the spins. $\theta = 0°$ is along the direction of the applied current. The black line in the base Py layer simulation image is used to outline the position of the nanomagnets on the ASI top layer. The spin direction of the two ends curls away from the long axis. And the dipolar field generated by the non-colinear spin structure within each nano-magnet of the ASI top layer will affect the local spin alignments in the Py base layer.

showing excellent performance for device application (see Supplementary Table 1).

To understand the transport behavior of the ASI, one needs to perform a model calculation counting the AMR and GMR effects for each nano-magnet, especially the influence of the off-axis spin structures at the ends of each nano-magnet. In order to calculate the effective resistance, we further simplify the 3D ASI structure as a 2D continuous media whose resistivity is contributed from either GMR or AMR. The current density distribution **J** in the ASI structure is governed by the Ohm's law **E** = $\rho$**J** where **E** is the local electric field and $\rho$ is the spatially dependent resistivity. For the region where tri-layer structure locates, the resistivity is considered to be contributed by GMR effect with the form

$$\rho_{GMR} = \rho_P + (\rho_{AP} - \rho_P)\cos\varphi \tag{1}$$

where $\varphi$ is relative angle of the average magnetization between the top layer and bottom layer, and $\rho_P(\rho_{AP})$ are resistivity for parallel (antiparallel) state for the tri-layer structure whose value is detected via experiment on a tri-layer single wire device (see Supplementary Fig. S3A). For the rest region of ASI structure (Py base layer), the resistivity is considered to be from AMR effect with the form

$$\rho_{AMR} = \rho_\perp + (\rho_\parallel - \rho_\perp)\cos^2\alpha \tag{2}$$

where $\alpha$ is the angle between the local magnetization and the current, and $\rho_\parallel$ ($\rho_\perp$) is the resistivity for currents flowing parallel (perpendicular) to the magnetization direction, whose value is detected via experiment on a pure Py base film (see Supplementary Fig. S3B). Since an in-plane field along the current direction is applied before transport measurement, $\rho_{AMR} = \rho_\parallel$ for most of the Py base layer except for the transition region close to the ends of each nano-magnet (see Fig. 3). For each of the eight ASI spin state, the calculated resistance of the tri-layer structure is shown in Fig. 4c, which are distinguishable with similar relative value to that of experimental data in Fig. 4a. Our model calculation indicates that different ASI spin state is distinguishable using transport measurements due to the GMR effect in the tri-layer structure.

The capability to distinguish ASI structures using transport measurements allows us to explore the full potential of device applications of ASI[35]. In particular, the fact that a large number of spin states can be created and tuned by external magnetic field in ASI makes ASI an ideal platform for reservoir computing (RC)[36–38]. Figure 5 shows the RC setup and task performance using our tri-layer ASI. In-plane magnetic fields (-70 Oe) with changing angle between the current direction ($\theta$) are used as input signals, and the output signals are resistance readouts between electrodes shown in Fig. 5a. Here the current direction is fixed with the source (G7) and drain (G8) Au contacts covering the full sample width to ensure a uniform current injection. Three pairs of

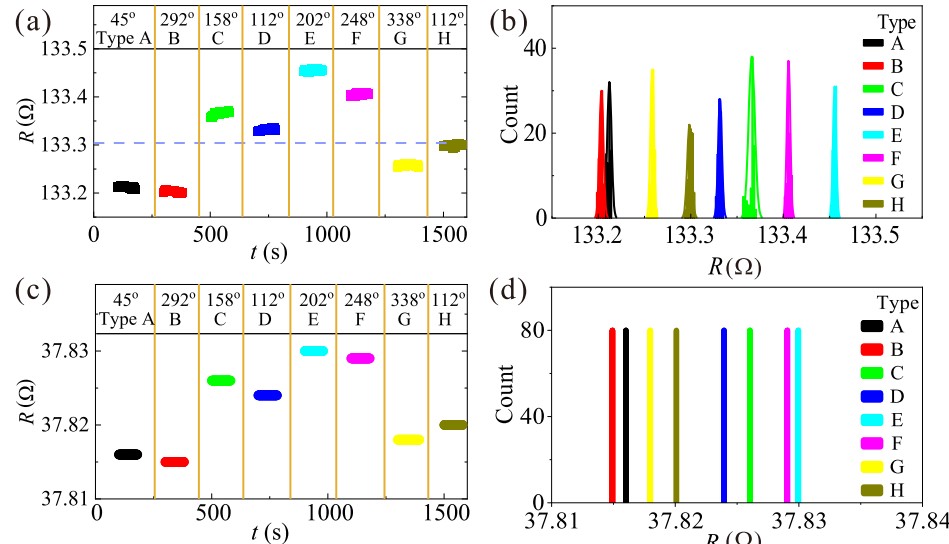

**Fig. 4 | Transport characterization of eight spin configurations.** All the resistance is measured at $\theta = 0°$ after the magnetic stimuli at certain angle with a small field to eliminate the AMR effect from most of the Py base layer. **a** Resistance of the eight long-range ordered spin configurations in the ASI tri-layer structure with the sequence angle of in-plane magnetic field at room temperature. The dash line represents the resistance value of the demagnetized state. **b** The separated eight peaks of the histogram of the acquired resistance data. **c** The calculated resistance of the eight long-range ordered spin configurations in the ASI tri-layer structure in model calculation. **d** The separated eight peaks of the histogram of the calculated resistance data.

electrodes are placed on two sides of the sample, which are marked as G1 to G6. The resistance value measured between any two of the six electrodes should reflect the combined effects of AMR, GMR, and planar Hall effects. Except for the eight spin states with long range order, the spin configuration of the ASI under an external field is spatially non-uniform[39,40] which can be sensitively measured by the resistance readouts between different electrode pairs. Figure 5c shows θ dependence for resistance measured between G1 and G3, i.e., $R_{1-3}$, which exhibits a desired nonlinear behavior for RC computing. $R$ vs. $θ$ curves measured between other pairs of electrodes exhibit nonlinear behavior with different characteristics indicating the spatially non-uniform distribution of spin configuration under an external magnetic field (see supplementary Fig. S8). Another important feature for RC system is fading memory. As show in Fig. 5d, we use the 8 long-range ordered spin configurations as the initial state to demonstrate fading memory behavior. $N$ number of fields are subsequently applied to the ASI, where the field angle $θ$ is randomly chosen from the eight angles used for preparing the eight long-range ordered spin states. While the resistance of the initial eight long-range ordered is clearly distinguishable ($N = 1$), with increasing $N$, resistance values, e.g., $R_{1-3}$, shows a clear tendency towards convergence to the same resistance value. This shows that the recent stimulus has a greater effect on the final spin configuration than the remote history.

Having verified that our ASI system meets the requirements of RC, we implement a reservoir computer based on the ASI device platform. Conventional RCs like the Echo State Networks consist of three parts: an input layer, a reservoir layer, and an output layer[35], as shown in Fig. 5a. Thereafter, an RC structure with only one physical node (Single Node RC)[41] but with far more virtual nodes from sampling the states of the physical node of the different moment has been proposed and proven to show great potentials in physically implementing RC on platforms like spin devices[42,43], memristors[44,45], and traditional analog circuits. Using our RC system, we execute predictions of two different time-series including Sunspot processing prediction[46,47] and performance on Mackey-Glass prediction[36,44]. The detailed RC computing is performed in the following steps: (1) We process the original input data series through a time-multiplexing process, multiplying the series with a mask matrix and converting the result of the multiplication into a train of step signals[48]. Every frame of the input signal can generate a

pulse train with total length $τ$; (2) The step signals are mapped to the range $[0, 2π]$, generating the field input to the ASI device where angle $θ$ acts as the variable; (3) The obtained set of resistance values with the field input are then processed using equation

$$M = M_0 + f(H) * (M_t - M_0), M_t = f(θ), \qquad (3)$$

where $M_0$ is the previous state, $M_t$ is the response value given by the nonlinear curve measured by the experiment after the input signals are mapped to the angle which is the character of the normalized resistance, and $f(H)$ is the coefficient related to the magnitude of the magnetic field. The generated data from this process are the output signals of the reservoir layer. We then integrate the RC output signals for each period $τ$ into the same frame of the reservoir states, and do ridge regression between the target series and frames of reservoir states to train the weight matrix of the readout layer. As shown in Fig. 5e, we test our RC with several benchmark tasks including Mackey-Glass prediction and Sunspot prediction, obtaining comparable results with the state-of-the-art results from other ASI-based RC[36,44,49]. The low NRMSE values are 0.1758 and 0.3293, respectively.

The introduction of Single Node RC and time-multiplexing process here can help when there is only limited dimension from the signal readout terminal, e.g., only one current readout terminal in memristors and nine resistance readout in the ASI device here, which is not adequate for the RC to map inputs into state space of higher dimensions. Also, the Single Node RC involves only a few devices in the implementation of RC, much fewer than the requirements of tradial ESNs, thus facilitating device-friendly application scenarios.

The way of combining spin information with electrical transport demonstrates that ASI-based reservoir computing is CMOS-compatible. The ASI reservoir itself can directly convert the spin configurations into the voltage signals as the reservoir states without an external load. The strong intrinsic coupling between the nanomagnets provides a random connection forming the reservoir network, beneficial for fabrication complexity. The richness of the reservoir states is one of the most important parameters in RC system. In previous works[34], different reservoir states were generated using the inherent device-to-device variations. Here, a large number of reservoir states are read in one single device by selecting multiple sets of electrodes. And the spin

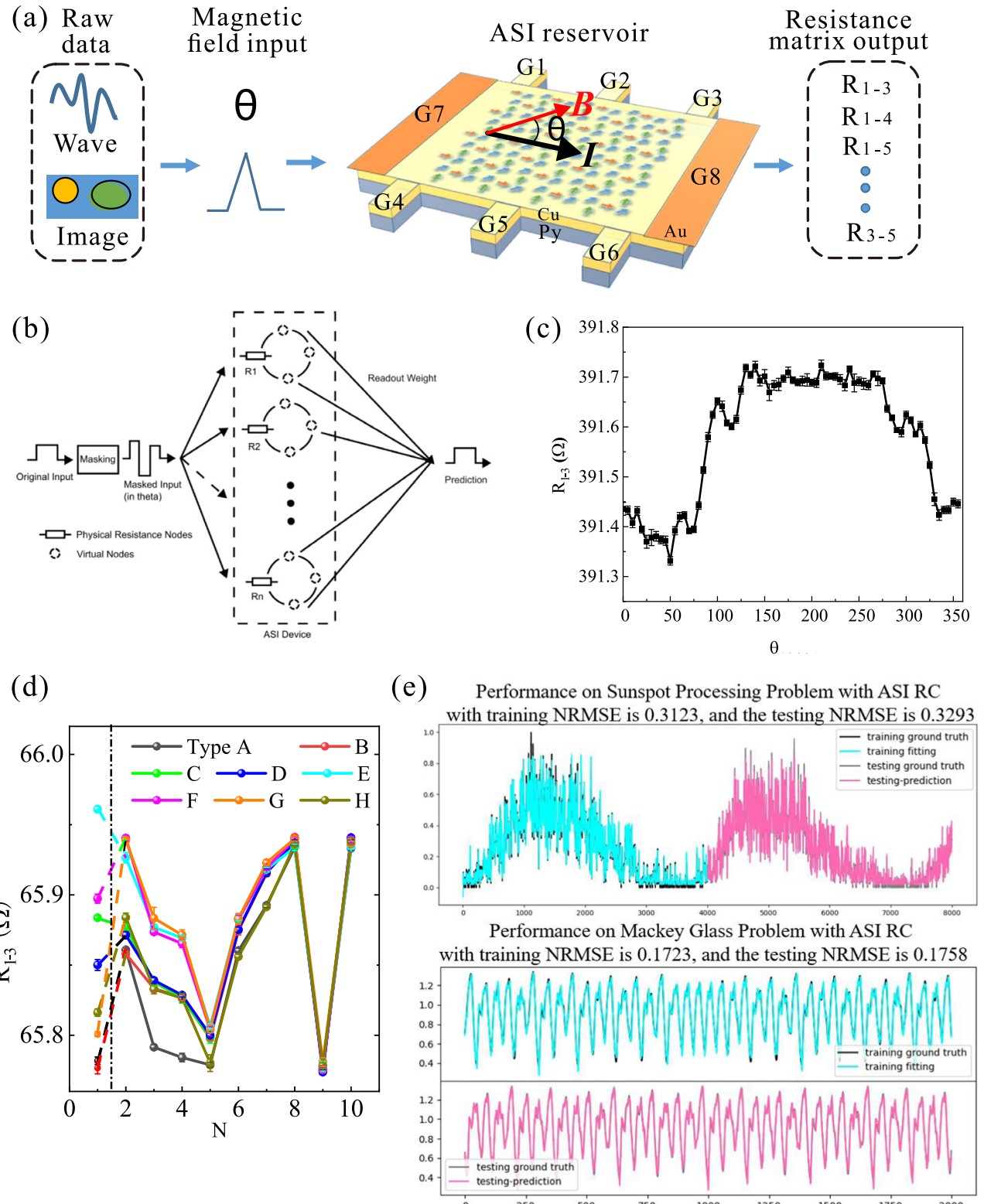

configurations in ASI are nonvolatile and at the same time they gradually 'forget' the previous states under external stimuli. In contrast to the dynamic memristor-based reservoir computing[48], the characteristic time of the short-term memory in ASI-based reservoir under a single excitation tends to be infinite. In any case, each virtual nodes are nonlinearly coupled with their neighbors in the mask process. Also, there is no long-term degradation under repeated stimulations in ASI devices as the spin acts as the information carrier. The endurance performance of memristor was well studied that a series of resistors can dramatically

improve the endurance of the TaO$_x$ memristor to $10^6$ cycles[50]. In the method of transport characteristic of ASI, a small current is applied to produce the voltage output. ASI can be stimulated indefinitely without degradation giving an excellent endurance performance.

In summary, we have demonstrated the capability to characterize ASI using transport measurements. By using the ASI structure to construct a GMR tri-layer, the combined GMR and AMR effects allow one to distinguish the resistance of different spin configurations in the ASI structure. We use the ASI system as a reservoir to complete the task

**Fig. 5 | The performance of reservoir computing based on the tri-layer ASI system. a** The concept of the ASI-based reservoir computing device. The device can be divided into three parts: the input, the reservoir, and the output. The input signals are encoded into the angle of the in-plane magnetic fields which applies globally on the ASI. The state evolution of ASI can be read out in situ via the transport measurement technique. Resistance matrix can be detected within each local segment as output, and fed into a following artificial neural network for further training and inference. **b** Structure for the ASI-based time-multiplexing RC. The original input data series is multiplexed to generate another data series where signal changes of higher frequencies are generated and linearly mapped to the range of the field angle. The resistance states between different pairs of electrodes are read out under the stimuli of the external magnetic field. The readout weight matrix is trained simply through ridge regression. **c** The nonlinear behavior of the ASI device used for the physical implementation of the RC system. We obtained the nonlinear curves of the input (angle of the in-plane magnetic field $\theta$ correspond to the current) and output signals (resistance) of 9 pairs of different electrodes. We map the input values linearly to the angle of the magnetic field (0–360 degrees) in a plane. The resistance of the ASI reservoir to the nonlinear response of all input signals was measured in a reproducible chaotic initial state. **d** ASI-based reservoir computing device has fading memory. The initial states are prepared with eight long-range-ordered spin configurations Type A to Type H. Then the same 9 input signals are given to eight different long-range-ordered spin configurations. The output resistances finally converge to the same behavior. The magnetic field applied is all about 70 Oe. **e** The performance on different benchmark tasks Sunspot processing prediction and Mackey-Glass prediction with ASI RC. We use half the data for training and half for testing. NRMSE is used to measure the classification error.

of time series fitting with high accuracy. Such a capability paves the way for device applications of ASI, whose rich spin states and especially the corresponding dynamic response to input signal are ideal for applications including RC, inverse-designed magnonic and multilevel data storage devices.

## Methods

### Sample fabrication

The heterostructure containing the $Ni_{80}Fe_{20}$ (Py) (6 nm)/Cu (4 nm)/Py (16 nm) films were prepared by the electron beam evaporation a rate of 0.2 angstrom per second. And tri-axial ASI was patterned by a combination of electron beam lithography and an argon ion milling process. We employed a lift-off technique using double layers of methyl methacrylates (MMA) and polymethyl methacrylates (PMMA). In the argon ion milling process we used 5 nm thick Al film for mask. The process yielded nanometer-scale Py islands with dimensions of 470 nm long, 170 nm wide and 16 nm thick on the Cu/Py film. And then 10 nm $Al_2O_3$ was used for prevent the whole sample for oxidation. The sample was then patterned into 300 μm (width) × 800 μm (length) area for longitudinal resistance measurement and 100 μm (width) × 200 μm (length) area for reservoir computing measurement. All the transport measurement was observed at room temperature 290 K with standard four-probe method. And the applied current was fixed at 20 μA along the horizontal sublattice avoiding the huge heating effect. The external magnetic field was fixed along the in-plane direction. The sample was rotated bringing a relative angle between the applied current and the magnetic field.

### MFM image

Magnetic force microscopy (MFM) imaging was conducted in a custom-designed MFM system using a commercial MFM probe (Point Probe Plus Magnetic Force Microscopy; NANOSENSORS). All the images were obtained from the tri-layer ASI device. All MFM images were obtained in zero fields.

### COMSOL simulation

We perform micromagnetic simulation using the COMSOL Multiphysics code[30] with the home-made Micromagnetics Module (http://www.physics.fudan.edu.cn/tps/people/jxiao/comsol-module-for-micromagn.html). We obtain eight different spin configurations of the nanomagnets in the ASI top layer as shown in Fig. 3, which correspond well to the eight ASI spin states determined by MFM. The simulations were performed in the smallest repeated cell with eight nano-magnets in the system under the periodic boundary condition in plane. We also take the under layers Cu (4 nm) and Py (6 nm) into the simulation. Here typical micromagnetic parameters for Py base layer and ASI array were used: saturation magnetization, $M_S = 860 \times 10^3$ A/m, exchange stiffness, $A = 13 \times 10^{-12}$ J/m, and zero magnetocrystalline anisotropy. The Gilbert damping parameter of $\alpha = 0.02$ allowed exploration of the energy landscape of the system[51]. Because the Py films were thin enough, the magnetization was almost constant in the thickness direction. When we calculated the resistance, we choosen the magnetization on the cross-section at the half height.

The calculation of effective resistance is also performed by COMSOL Multiphysics using AC/DC module. Because the Py base film and ASI array were thin enough, the magnetization was almost constant in the thickness direction. In each area, we choose the magnetization on the cross-section at half height and then calculate the resistance using the 2D model.

## Data availability

All raw and derived data used to support the findings of this work are available from the authors on request.

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

## Acknowledgements

This work was supported by the National Key Research Program of China (2020YFA0309100), the National Natural Science Foundation of China (11991062, 12074075, 12074073, 12074071, 11804052, 12074080), Shanghai Municipal Science and Technology Major Project (2019SHZDZX01), Shanghai Municipal Natural Science Foundation (20501130600, 22ZR1407400), Shanghai Science and Technology Committee Rising-Star Program (19QA1401000).

## Author contributions

J.S. and W.H. conceived the project and constructed the research frame. W.H. prepare the ASI arrays and built the test hardware. Q.L., H.Z., Y.S., and D.Z. performed the MFM measurements. Y.L., W.Y., J.X., and L.Y. analyzed the micromagnetic simulation results. Z.Z., X.Z., and Q.L. completed RC simulations. J.S., W.H., H.G., and W.W. analyzed the experimental data and simulation results. All authors discussed the results and implications and commented on the manuscript at all stages.

## Competing interests

The authors declare no competing interests.
