## [Peer Review File · Nature Communications]

Reviewers' Comments:

Reviewer #1:

Remarks to the Author:

I appreciate that the Authors have made strides by now providing a demonstration that the ASI system that they fabricated can be used as a reservoir computer. Given the concerns that I had in my original review, I do agree with the Authors that these demonstrations are needed for the paper to merit publication outside of a more specialized/technical journal.

In light of these improvements and additional experiments that were added to the paper, I do think that the novelty of the work has been made clearer, and I am willing to recommend publication for *Nature Communications*. However, I do have some suggestions that I think the Authors need to address before final acceptance.

1. The new paragraph between lines 183-206 needs some work. I would suggest that the Authors add some references regarding the time-series prediction studies that they added. Other examples of RC experiments that used the "Sun Dot" and Mackey-Glass time-series are probably useful references that can be added. Furthermore, I was unable to find in the literature other studies that used the "Sun Dot" time-series. Is this a non-standard name?
2. There is a typo in the added paragraph where the Authors refer to Figure R2 and not Figure 5.

Reviewer #2:

Remarks to the Author:

I read with interest the manuscript "Distinguishing Artificial Spin Ice States using Magnetoresistance

Effect for Neuromorphic Computing" by Dr. Hu and collaborators.

The work tackles one of the interesting open problems of exploring efficiently the latent space of a spin ice material via resistance measurements.

I believe that this is important work for the spin ice community. Currently, we can only perform indirect or destructive measurements in spin ice materials. The first paper introducing neuromorphic computing was Gartside et al (reference 38), but these were done via FMR measurement, and only via vortex fingerprinting. The methodology employed by the authors, and suggested previously in the literature, will be an important tool in future publications on spin ice. I am thus very optimistic about the manuscript, and I think it should be eventually published in *Nature Communications*.

The relevant figure is Figure 4. This is essentially a "magnetic resistograph", similar to spectroscopic measurements.

I find that this is a fairly appealing result, although I doubt how it could be used for larger arrays when degeneracy will be present. This should be certainly mentioned in the manuscript, but it is not a limiting factor in the provided implementation.

Some minor points: I believe that the authors fail to cite relevant references. These were correctly pointed out by another referee and were not cited in the resubmitted manuscript.

Such memory effects were first observed in reference 39, and the theory for the single-node effects introduced in Chern (not cited):

*) Chern Magnetotransport in artificial Kagome spin ice Phys. Rev. Appl. 8 064006 (2017)

for the specific case of 3-island junctions, and a more general theory developed in reference 40. Typical AMR measurements were done more recently in (not cited)

Fonseca et al, Memristive Effects in Nanopatterned Permalloy Kagomé Array, Phys. Rev. Applied 18, 014070 2022,

which was suggested by another referee, and whose measurement resistance change values are consistent with the experimentally observed changes in the paper by the authors.

Regarding the RC result, what the authors provide is a valuable proof of concept. However, Supplementary material Figures S8 and S9 show that this physical system has the right properties for RC implementation.

I thus have only very minor additional comments about the manuscript compared to the other referees: I think this is an important paper, with a very relevant experimental setup that will enable future neuromorphic applications. I thus endorse the publication in Nature Communications, provided that the comments above are added to the paper.

In this letter we provide a point-to-point response to the reviewers' comments. The reviewer's original comments are shown by blue italic characters. The authors' responses are shown by black normal characters.

Reviewer #1 (Remarks to the Author):

I appreciate that the Authors have made strides by now providing a demonstration that the ASI system that they fabricated can be used as a reservoir computer. Given the concerns that I had in my original review, I do agree with the Authors that these demonstrations are needed for the paper to merit publication outside of a more specialized/technical journal.

In light of these improvements and additional experiments that were added to the paper, I do think that the novelty of the work has been made clearer, and I am willing to recommend publication for Nature Communications. However, I do have some suggestions that I think the Authors need to address before final acceptance.

Thank you very much for taking time to review our manuscript and support for publication.

1. The new paragraph between lines 183-206 needs some work. I would suggest that the Authors add some references regarding the time-series prediction studies that they added. Other examples of RC experiments that used the "Sun Dot" and Mackey-Glass time-series are probably useful references that can be added. Furthermore, I was unable to find in the literature other studies that used the "Sun Dot" time-series. Is this a non-standard name?

Thanks for the suggestions. We apologize for the misleading phrase "Sun Dot", which is corrected as "Sunspot" in the revised manuscript. We have added references 43,48-

50 that used the Sunspot and Mackey-Glass time-series as the standard tasks in RC experiments.

2. There is a typo in the added paragraph where the Authors refer to Figure R2 and not Figure 5.

We thank the reviewer for pointing this out. We have corrected the mistake.

Reviewer #2 (Remarks to the Author):

I read with interest the manuscript "Distinguishing Artificial Spin Ice States using Magnetoresistance Effect for Neuromorphic Computing" by Dr. Hu and collaborators. The work tackles one of the interesting open problems of exploring efficiently the latent space of a spin ice material via resistance measurements.

I believe that this is important work for the spin ice community. Currently, we can only perform indirect or destructive measurements in spin ice materials. The first paper introducing neuromorphic computing was Gartside et al (reference 38), but these were done via FMR measurement, and only via vortex fingerprinting. The methodology employed by the authors, and suggested previously in the literature, will be an important tool in future publications on spin ice. I am thus very optimistic about the manuscript, and I think it should be eventually published in Nature Communications.

We thank the reviewer for acknowledging the novelty of our work and his/her support of publication in Nature Communications.

The relevant figure is Figure 4. This is essentially a "magnetic resistograph", similar to spectroscopic measurements. I find that this is a fairly appealing result, although I doubt how it could be used for larger arrays when degeneracy will be present. This should be certainly mentioned in the manuscript, but it is not a limiting factor in the

provided implementation.

The reviewer has raised an important issue regarding the measurements of a large array with degenerate states. While the number of ordered spin states (8) remains unchanged no matter how large the array is, the number of spin states with non-ordered spin configurations becomes very large with increasing size of the array. In this case, obtaining the “magnetic resistograph” by a single pair of electrodes becomes virtually impossible. Instead, we can obtain a complex “magnetic resistograph” based on the measured resistance matrix from an increased number of electrodes around the array, as we did in the RC experiment.

Some minor points: I believe that the authors fail to cite relevant references. These were correctly pointed out by another referee and were not cited in the resubmitted manuscript.

Such memory effects were first observed in reference 39, and the theory for the single-node effects introduced in Chern (not cited):

**) Chern Magnetotransport in artificial Kagome spin ice Phys. Rev. Appl. 8 064006 (2017) for the specific case of 3-island junctions, and a more general theory developed in reference 40. Typical AMR measurements were done more recently in (not cited) Fonseca et al, Memristive Effects in Nanopatterned Permalloy Kagomé Array, Phys. Rev. Applied 18, 014070 2022, which was suggested by another referee, and whose measurement resistance change values are consistent with the experimentally observed changes in the paper by the authors.*

The references mentioned are highly relevant and are added as references 35 and 36 in our revised paper. We will add another reference as No.43 which specifically discussed the Single Node issue in RC (Nature Commun. 2, 1-6 (2011)).

Regarding the RC result, what the authors provide is a valuable proof of concept.

However, Supplementary material Figures S8 and S9 show that this physical system has the right properties for RC implementation.

We fully agree with the reviewer's statement that our ASI system has the right properties for RC implementation (Supplementary Figures S8 and S9). Moreover, as shown in Fig.5 we have successfully executed two real tasks based on the ASI-RC system.

I thus have only very minor additional comments about the manuscript compared to the other referees: I think this is an important paper, with a very relevant experimental setup that will enable future neuromorphic applications. I thus endorse the publication in Nature Communications, provided that the comments above are added to the paper.

We thank the reviewer again for these valuable comments, which have been fully addressed in the revised paper.

Below is a list of the main changes to the manuscript:

- (1) We have added more references (ref.35,36, 48-50).
- (2) We have corrected the typos “Sunspot” and “Fig. 5(e)”
- (3) We rearranged the order of work units.